# Assessing the Level of Knowledge, Beliefs and Acceptance of HPV Vaccine: A Cross-Sectional Study in Romania

**DOI:** 10.3390/ijerph19116939

**Published:** 2022-06-06

**Authors:** Toader Septimiu Voidăzan, Mihaela Alexandra Budianu, Florin Francisc Rozsnyai, Zsolt Kovacs, Cosmina Cristina Uzun, Nicoleta Neagu

**Affiliations:** 1Department of Epidemiology, George Emil Palade University of Medicine, Pharmacy, Science, and Technology of Targu Mures, 38 Gheorghe Marinescu Street, 540139 Targu Mures, Romania; septimiu.voidazan@umfst.ro (T.S.V.); neagunicoleta92@yahoo.com (N.N.); 2Department of Obstetrics Gynecology, George Emil Palade University of Medicine, Pharmacy, Science, and Technology of Târgu Mureș, 38 Gheorghe Marinescu Street, 540139 Targu Mures, Romania; florin.rozsnyai@umfst.ro; 3Department of Biochemistry, Environmental Chemistry, George Emil Palade University of Medicine, Pharmacy, Science, and Technology of Târgu Mureș, 38 Gheorghe Marinescu Street, 540139 Targu Mures, Romania; zsolt.kovacs@umfst.ro (Z.K.); cosmina20uzun@gmail.com (C.C.U.)

**Keywords:** HPV vaccination, HPV infection, sources of information, internet, patient education

## Abstract

(1) Background: The infection with Human papilloma virus (HPV) is the most common sexually transmitted infection and it has been associated with cervical cancer (CC) in 99.7% of the cases. In Romania, CC is the second most common, with incidence (22.6%_000_) and mortality rates (9.6%_000_) three times higher than any other European country. Our aim was to assess the level of knowledge regarding HPV infection among parents, highschool students, medical students and doctors, with an emphasis on their main source of information—the Internet. (2) Methods: We applied five questionnaires to six categories of respondents: parents of pupils in the 6th–8th grades, medical students, doctors, boys in the 11th–12th grades, girls in the 11th–12th grades and their mothers. (3) Results: We included a total of 3108 respondents. 83.83% of all respondents had known about HPV infection. The level of information about HPV infection and vaccination was either satisfactory, poor or very poor. Their main source of information varied depending on the respondent profile and professional activity. Medical students were informed by doctors and healthcare professionals (53.0%), doctors gathered their information from books, journals and specialized brochures (61.6%). For the other categories of respondents, the Internet was the main source of information. Most respondents answered that doctors and healthcare professionals should provide information on HPV infection and vaccination, but very few of them actually seeked information from their general practitioner. (4) Conclusions: Population adherence to the appropriate preventative programs, as well as relevant information disseminated by the medical staff are key elements towards reducing the risk of HPV-associated cancers. An important role could also be played by schools, where teachers and school doctors could provide relevant information on the general aspects of HPV infection. Additionally, sex education classes and parent-teacher meetings should cover the main characteristics of HPV infection and what preventative measures can be employed against it.

## 1. Introduction

The infection with Human papillomavirus (HPV) is the most common sexually transmitted infection worldwide and it has been associated with cervical cancer (CC) in 99.7% of the cases, thus being considered an important etiological agent [1,2]. So far, more than 200 HPV types have been indexed, 15 of which being classified as high-risk (HR) (16, 18, 31, 33, 35, 39, 45, 51, 52, 56, 58, 59, 68, 73, 82), three as probable high risk (26, 53, 66) and 12 as low risk (6, 11, 40, 42, 43, 44, 54, 61, 70, 72, 81, CP6108). HPV16 and 18 are the most carcinogenic types: HPV16 has been correlated with 50–60% CC, HPV18 with 10–15% CC and the remaining HR-HPV types have been implicated in 25–40% of CC [3,4]. Additionally, high-risk HPV types can cause anal, penile and oropharyngeal cancer, while low-risk genotypes are responsible for genital warts and respiratory papillomatosis [5,6]. However, most HPV infections are transient, due to the immune system’s capacity to eliminate the virus [7].

CC is the fourth most common type of cancer among women worldwide and in Romania it is the second most common, with incidence rates (22.6%_000_) and mortality rates (9.6%_000_) three times higher than in any other European country. Current data shows that every year, 3380 women in Romania are diagnosed with CC and 1805 of them die [8,9,10]. Additionally, the incidence of HPV infection among women aged 18–59 years is 40% for all HPV types and 20% for HR-HPV types [11].

In Romania, CC screening is performed by Babeș-Papanicolau test (Pap test), which was implemented in 2012 and it addresses women aged 25–64 years who have no symptoms suggestive of CC, a previous diagnosis of CC, or who had undergone total hysterectomy. The test is repeated every 3 years for women aged 25–49 years and every 5 years for women aged 50–64 years [12].

However, the lack of a national cancer registry is a major barrier to reducing CC screening in Romania [13]. Additionally, in Romania, HPV vaccination is not currently part of the National Vaccination Program. There was a vaccination campaign against HPV infection implemented in 2008 which targeted girls aged 10–11 years, but only 2.57% of the 110,000 eligible girls received the vaccine. The same campaign was resumed in 2009, this time targeting girls aged 12–14 years. Low HPV vaccine uptake rates after information campaigns were due to poor management, patient hesitancy and lack of involvement from the medical personnel [2].

Consequently, by the end of 2010, the programme was discontinued [2,14,15]. Another attempt at implementing another HPV vaccination campaign began in January 2020, when the Ministry of Health provided free HPV vaccine doses for girls aged 11–14 years [16]. In September 2021, the Romanian Ministry of Health announced that free HPV vaccination was extended to 18-year-old girls and that vaccine doses would become available for free, for girls aged 11–18 years, at the general practitioner’s office, on the basis of a waiting list schedule [17].

In Romania, low HPV vaccine uptake rates is a consequence of low adherence to screening programs for eligible women, lack of parental approval for HPV vaccination, as well as a general lack of education regarding HPV infection and the severity of its consequences [2].

Preferred sources of information that the Romanian population uses are mass-media (radio, television, press, etc.), healthcare professionals, medical specialty books and brochures, the Internet, parents, relatives and school teachers [18,19].

The Internet, with a special emphasis on social media platforms, can significantly impact personal awareness, knowledge, attitudes and beliefs regarding HPV infection and vaccination [20]. Dunn et al. [21] analyzed the impact social media websites like Twitter had on HPV vaccine uptake. They showed that information exposure on Twitter was linked to differences in uptake rates and that it was not influenced by socioeconomic factors: posts focused on conspiracy theories and vaccine safety were associated with lower vaccine uptake, while positive, scientifically-based topics had a weaker correlation with vaccine coverage.

Understanding the general population’s personal beliefs regarding HPV vaccination and assessing their level of knowledge on HPV as an etiological agent for CC are inextricably linked to development of an effective health policy supporting HPV vaccination and CC screening programs [22]. Population-based information programs have the potential of raising awareness by providing accurate information about HPV infection and its role in developing CC, as well as information regarding the safety and efficacy of HPV vaccine [22,23].

Given the severity of HPV infection complications, its potential for malignant transformation, its high prevalence and the lack of adherence of the Romanian population to CC screening and HPV vaccination programs, HPV is still a topic of national interest. In the digital age, where access to information is unrestricted, the Internet has become a reliable source of information, sometimes to the detriment of appropriate information sources, such as physicians and medical brochures. Our aim was to assess the level of knowledge about HPV infection among students, adolescents, mothers, parents, and physicians, as well as to reveal their main source of information.

## 2. Materials and Methods

We conducted a cross-sectional study in which we applied five questionnaires to six categories of respondents: parents of pupils in the 6th–8th grades, medical students, doctors, highschool boys in the 11th–12th grades, highschool girls in the 11th–12th grades and their mothers. Each of the five questionnaires mentioned the purpose of the study and included a data anonymity and confidentiality statement. These questionnaire-based studies were approved by The Ethics Committee of the University of Medicine and Pharmacy of Târgu Mureș (no. 55/13.05.2019). The questions were either open-ended, closed with ordered answers, closed with unordered answers, or binary. The respondents were given 15 min to complete the questionnaires. The questions were focused on the level of knowledge about CC, the preventative role of HPV vaccination, the attitude towards serious disease prevention and their preferred sources of information. The level of knowledge about HPV infection and vaccination was rated on a scale from 1 to 5: 1-very poor, 2-poor, 3-satisfactory, 4-good, 5-very good (Table 1). The characteristics of each questionnaire are presented below:The first questionnaire was self-applied to 918 parents of pupils in the 6th–8th grades and it consisted in 34 questions [2]. Three Middle Schools from Târgu Mureş City were randomly selected and had a total of 1180 the 6th–8th graders, representing approximately 10.0% of the 6th–8th graders in the city. The questionnaire included 34 questions.The second questionnaire was self-administered to 290 medical students in the 1st and 232 in the 6th year at the University of Medicine, Pharmacy, Science and Technology ‘George Emil Palade’ of Târgu Mureş and it included 55 questions [18]. They were evaluated comparatively in order to establish the difference in knowledge between the two groups.The third questionnaire was sent online to 1411 doctors from three different medical specialties: 1344 general practitioners, 42 pediatricians and 25 obstetricians/gynecologists. Only 640 responded: 580 general practitioners, 39 pediatricians and 21 gynecologists, with an overall response rate of 43.1% for general practitioners, 92.8% for pediatricians and 84% for gynecologists. It included 38 questions [19].The fourth questionnaire was distributed among 654 highschool students in the 11th–12th grades from two highschools in Făgăraș City. The questionnaires were partly different among boys and girls and they included 22 and 19 questions, respectively.The fifth questionnaire consisted in 29 questions and it was self-administered to 187 girls in the 11th–12th grades and their mothers. Three High Schools were randomly chosen, two of which from Târgu Mureș and one from Miercurea Ciuc.


### Statistical Analysis

Data was collected in Excel forms. Statistical analysis was performed using the Statistical Package for Social (SPSS, version 23, Chicago, IL, USA). Data were labelled as nominal or quantitative variables. Nominal variables were characterized by means of frequencies.

## 3. Results

We included a total of 3108 respondents, of which 918 were parents of pupils in the 6th–8th grades, 290 1st year and 232 6th year medical students, 640 doctors, 354 boys in the 11th–12th grades, 487 girls in the 11th–12th grades and 187 of their mothers.

### 3.1. Level of Knowledge

83.8% of all respondents had known about HPV infection, of which 85.8% parents of pupils in the 6th–8th grades, 89.7% medical students, 100% doctors, 70.4% boys in the 11th–12th grades, 74.3% girls in the 11th-12th grades and 82.8% of their mothers. The level of information about HPV infection was satisfactory for almost half of the doctors (47.3%), this level being identified in around one third of most respondents except for girls, of whom 32.4% and 24.8% had a poor and a very poor level of information, respectively. A good level of information was identified in 32.1% doctors and 28.9% medical students. The degree of information on HPV vaccination was satisfactory for 44.6% doctors, while for the other categories of respondents the poor level of information was predominant (Table 1).

Regarding the optimal age for HPV vaccine administration, 69.0% of the medical students, 72.3% doctors and 30.9% boys in the 11th–12th grades considered that optimal vaccination should occur between 12–14 years, while 44.5% of the parents of pupils in the 6th–8th grades and 50.6% of the mothers of high-school students in the 11th–12th grades believed that vaccination should be considered after the age of 18 years. 50.6% girls in the 11th–12th grades believed that vaccination should be administered before becoming sexually active.

The majority of respondents believed that HPV infection could not be transmitted if they used a condom: 60.8% of parents of pupils in the 6th–8th grades, 73% medical students, 92.9% doctors, 50.5% boys in the 11th–12th grades, 64.7% girls in the 11th–12th grades and 55.7% of their mothers. Medical students and doctors believed that HPV infection could be prevented by resuming sexual activity to one sexual partner or by HPV vaccination, with a frequency of 68.1% versus 70.5% and 63.7% versus 86.6%, respectively. Also, 49.5% of girls in the 11th–12th grades and 52.3% of their mothers believed that vaccination was an effective measure to prevent HPV infection.

### 3.2. Information Sources

The main source of information on HPV infection varied depending on the profile of the respondent and their professional activity. Medical students preferred to be informed by doctors and healthcare professionals (53.0%), as well as by using the Internet (47.4%). Doctors would rather get updates from books, journals and medical brochures (61.6%) and from the Internet (46.4%). For the other categories of respondents, the Internet was the main source of information: 42.9% parents of pupils in the 6th–8th grades, 58.6% boys in the 11th–12th grades, 78.3% girls in the 11th–12th grades and 55.7% of their mothers.

When asked who would be more suitable to provide information about HPV infection or vaccination, the majority of the respondents preferred doctors or healthcare professionals (Table 2). However, very few respondents seeked information regarding HPV infection and vaccination from their general practitioner: 26.4% of parents of pupils in the 6th–8th grades, 11.9% medical students, 13.7% boys in the 11th–12th grades, 6.7% girls in the 11th–12th grades and 18.4% of their mothers.

## 4. Discussion

Human Papilloma Virus is estimated to affect approximately 50–80% of sexually active women and men throughout their lifetime. Studies have shown that infection with HPV types 16, 18, 31, 33, 35, 45, 52,58 have an increased risk of malignant transformation in both males and females [24]. In Europe, HPV infection is considered to be a woman’s health issue and most countries offer vaccination against this virus exclusively to women, although HPV has also been associated with cancers in male patients: anal, penile, oral and oropharyngeal cancers [7].

According to the World Health Organization, most countries recommend vaccination against HPV at the age of 9–12 years, the main target group being adolescent girls. This outdated focus on women vaccination might explain the gender gap in HPV education [25]. Hence, it is recommended that doctors advocate for HPV vaccination in both girls and boys between 11 and 14 years of age. Moreover, HPV vaccination could become part of the routinely administered vaccines [26].

Until 2021, in Romania, HPV vaccination was administered free of charge exclusively to girls aged 11–14 years. The Ministry of Health has since extended the target group to girls aged 14–18 years, which is in line with the Epidemiology and Oncology Commissions of the Ministry of Health and the National Institute of Public Health recommendations [17].

From the records of our study, 69.0% medical students and 72.3% doctors answered that the optimal vaccination age against HPV infection was between 12 and 14 years. The majority of the girls surveyed (50.6%) considered vaccination to be appropriate before becoming sexually active and 44.5% parents of pupils in the 6th–8th grades and 50.6% of the highschool student’ mothers answered that vaccination should be administered after the age of 18 years.

So far, studies at European level have shown that adolescents have a low level of knowledge regarding HPV infection and vaccination [27]. One meta-analysis estimated that adolescent girls were more likely to know about HPV infection (51.7% versus 29.8%) and vaccination (45.1% versus 18.9%) than boys. In our study, however, after applying the questionnaires to highschool students in the 11th–12th grades, we found that the majority of the boys (34.2%) considered themselves to have a satisfactory level of information about HPV infection, as compared to girls, who had a poor level of information (32.4%), while, when asked about HPV vaccination, 33.4% of the boys and 45.7% girls had a poor level of knowledge. These results show a very low level of knowledge among students and their parents regarding the clinical signs, methods of transmission, infection site, infection consequences, risk factors and prophylaxis of HPV infection. Similarly, Smolarczyk et al. [28] demonstrated an insufficient level of knowledge about HPV virus and HPV vaccine among parents, which was variable with gender, age and education, irrespective of the place of residence and the number of children they had. Additionally, a low level of knowledge about HPV virus and HPV vaccine was found among physicians, regardless of their gender, age, or specialization [29].

The importance of information sources on HPV infection has been demonstrated through numerous studies which showed that individuals gathering information from the Internet had a higher level of knowledge and were more likely to accept HPV vaccine [30,31]. Healthcare professionals shared medical information with their patients and the general public using social media. Also parents, teenagers and young adults used social media as a source of medical information [32,33]. However, there were also negative posts making inaccurate statements, addressing vaccine side effects and promoting conspiracy theories involving distrust of pharmaceutical companies, physicians and the government [15,21,34,35]. Positive and pro-vaccine social media posts included information regarding prevention, protection, efficacy and safety data [21,35].

Social networks (Facebook, Twitter and YouTube) have become an increasingly popular source of medical information for people around the world. They democratize the ability of both physicians and non-healthcare professionals to share their experiences and opinions on medical issues to a wide audience, irrespective of the medical accuracy of their information. The alarmingly rapid dissemination of inaccurate, medically biased information about health issues on social networks poses a great threat to the general population, who cannot discern between authentic and false information [20,36,37,38,39]. The Internet is an increasingly popular tool among parents who research HPV vaccination. Surveys of teenagers’ parents have shown that parents who accessed information on HPV vaccine were more knowledgeable about HPV and had a more positive attitudes towards vaccination.

The medical advice given to parents plays an important part in the decision to vaccinate their children, especially information regarding the safety and efficacy of the vaccine [40,41]. Also, information gathered from mass-media has an important role in the parents’ decision [42]. Unfortunately, anti-vaccination campaigns continue to fuel parents’ mistrust and these are largely promoted through social media platforms and television. According to our study, mothers who refuse vaccinating their daughters against HPV admit having poor information about HPV vaccine, believe that the vaccine had been insufficiently tested, or prefer to have ample discussions with their daughters and explain how to protect themselfes against HPV infection.

Considering that Romania ranks second in Europe with a critically high number of deaths from CC, it can be argued that there is a major lack of education about HPV infection, its role in CC development and HPV vaccination as a preventative method, which is in agreement with our findings. Additionally, information campaigns exclusively targeting women has lead to a complete disregard of representatives of the opposite sex. Lack of adherence to screening programs and parental opposition indicate low levels of education about HPV infection and its long-term implications. This makes HPV infection a matter of national interest, with a major impact on the morbidity and mortality of both Romanian men and women.

## 5. Limitations

One major limitation of our study is given by the assessment of the population from one country alone, without prior application of a probabilistic sampling formula, thus influencing the generalization of the results. However, the large sample size, high response rates and the heterogeneity of socio-demographic background of the respondents tackle this vulnerability. Another limitation derives from the completion method of the questionnaires, which were self-administered and might bring into question the level of comprehension of the respondents and thus, the reliability of their answers. We reduced this limitation by excluding both incomplete and invalid questionnaires, after thorough internal validity assessments. The questionnaires applied to physicians had unsatisfactory low response rates, of 43.1% for family doctors, which might be explained by either the online completion method or the reluctance of the respondents, even though anonymity and confidentiality were respected. The questions we applied were rather subjective and did not address specific information regarding HPV infection and HPV vaccination.

## 6. Conclusions

In our study, most respondents gathered their information about HPV infection and vaccination from Internet sources and their level of knowledge was self-assessed as either satisfactory or poor. Very few respondents seeked information from their general practitioner or HPV specialist, at the same time considering that accurate information about HPV should be provided by physicians and healthcare professionals. In order to reduce the risk of HPV-associated cancers, populational adherence to screening programs and dissemination of relevant information by medical staff are key elements.

Vaccination information campaigns and screening programs are effective ways to reduce the incidence, mortality and morbidity associated with HPV infection. Teenager and young adult education on HPV could make them more sexually responsible, which might also determine them to get vaccinated. An important role could also be played by schools, where biology teachers, class masters and school doctors could provide relevant information on the general aspects of HPV infection. Additionally, sex education classes could be held by a mixed team of medical specialists and psychologists, covering sexual health topics like HPV infection and vaccination, as well as other sexually transmitted infections prevention. Equally important are the parent-teacher meetings discussing the importance of HPV vaccination and any other concerns that parents might have regarding the vaccine.

## Figures and Tables

**Table 1 ijerph-19-06939-t001:** Level of knowledge on HPV infection and vaccination.

Questions	Answers	Survey 1 Parents No = 918	Survey 2 Students No = 522	Survey 3 Doctors No = 640	Survey 4 Boys No = 354	Survey 4 + 5 Girls No = 487	Survey 5 Mothers No = 187
Have you ever heard of HPV infection (human papilloma virus)?	○ Yes	85.8	89.7	100.0	70.4	74.3	82.8
	○ No	14.2	10.3	0.0	29.6	25.7	17.2
How would you describe your level of knowledge on HPV infection?	○ Very poor	11.1	12.6	0.9	15.6	24.8	14.9
	○ Poor	26.0	20.9	15.2	28.5	**32.4**	17.2
	○ Satisfactory	**31.5**	**30.1**	**47.3**	**34.2**	18.1	**35.6**
	○ Good	24.4	28.9	32.1	11.9	21.0	24.1
	○ Very good	7.0	7.5	4.5	9.8	3.7	8.2
How would you describe your level of knowledge on HPV vaccination?	○ Very poor	14.8	20.5	5.4	20.2	31.4	17.2
	○ Poor	**31.7**	**32.2**	22.3	**33.4**	**45.7**	**33.3**
	○ Satisfactory	30.6	27.0	**44.6**	25.7	16.2	24.1
	○ Good	18.4	15.5	24.1	15.3	4.8	19.5
	○ Very good	4.5	4.8	3.6	5.4	1.9	5.9
Which do you consider to be the optimal age for HPV vaccine administration?	○ 0–11 years	8.1	9.0	17.0	19.0	5.8	6.9
	○ 12–14 years	36.5	**69.0**	**72.3**	**30.9**	21.0	24.1
	○ 15–17 years	6.0	4.6	5.4	14.5	22.9	16.1
	○ After the age of 18 years	**44.5**	14.2	0.9	13.5	14.9	**50.6**
	○ Before sexual activity inception	2.5	1.8	27.7	20.5	**50.6**	1.1
	○ I don’t know	2.4	1.4	0.9	1.6	2.5	1.1
How do you think you can prevent HPV infection?	○ Having one sexual partner	37.0	68.1	70.5	14.1	29.5	39.8
	○ Vaccination	48.1	63.7	86.6	40.1	49.5	52.3
	○ Using a condom	**60.8**	**73.0**	**92.9**	**50.5**	**64.7**	**55.7**
	○ Using oral contraceptives	74.6	3.3	2.7	6.5	4.76	3.4
	○ Local hygiene with soap and water after intercourse	1.3	39.8	33.9	4.5	22.8	6.8
	○ Late sexual activity inception	64.5	18.9	30.4	11.9	8.57	11.4
	○ Babeș-Papanicolau testing	94.2	58.9	40.2	33.21	24.7	36.4

**Table 2 ijerph-19-06939-t002:** Information sources for HPV infection and vaccination.

Questions	Answers	Survey 1 Parents No = 918	Survey 2 Students No = 522	Survey 3 Doctors No = 640	Survey 4 Boys No = 354	Survey 4 + 5 Girls No = 487	Survey 5 Mothers No = 187
Which are your main sources of information on HPV infection?	○ Doctors, healthcare professionals	39.1	53.0	31.3	9.7	24.7	33.0
	○ Parents, relatives	3.0	10.1	NA	15.3	17.2	4.5
	○ Friends, acquaintances	11.7	8.0	NA	16.8	7.6	6.8
	○ Books, journals, medical brochures	22.0	31.9	**61.6**	33.4	15.2	40.9
	○ Newspapers, radio	32.4	17.6	29.5	0.74	4.76	13.6
	○ Internet	**42.9**	47.4	46.4	**58.6**	**72.3**	**55.7**
In your opinion, who do you think is most qualified to give most information on HPV infection and vaccination?	○ Doctors, healthcare professionals	**85.5**	**90.4**	**61.4**	**49.1**	**54.7**	**44.5**
	○ Books, journals, medical brochures	5.1	29.6	37.5	14.3	16.2	17.3
	○ University teachers	0.9	43.8	15.6	15.8	7.6	7.6
	○ Newspapers, radio, TV	15.8	33.3	52.6	38.4	40.2	38.2
	○ Friends, acquaintances	17.6	6.7	9.5	10.6	14.6	17.3
	○ Parents, relatives	22.5	18.9	6.4	28.6	12.3	8.3
Have you received any information on HPV infection and vaccination from your general practitioner or HPV specialist?	○ Yes	46.2	32.3	NA	9.6	13.3	21.8
	○ No	**53.8**	**67.7**		**90.4**	**86.7**	**78.2**
Have you requested any information on HPV infection and vaccination from your general practitioner or HPV specialist?	○ Yes	26.4	11.9	NA	13.7	6.7	18.4
	○ No	**73.6**	**88.1**		**86.3**	**93.3**	**81.6**

## Data Availability

Not applicable.

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
