# Peer review of "Assessing the Level of Knowledge, Beliefs and Acceptance of HPV Vaccine: A Cross-Sectional Study in Romania"

_ijerph, 2022, doi:10.3390/ijerph19116939_

Round 1

Reviewer 1 Report

The present manuscript discusses the level of HPV knowledge among the parents, high school students, medical students, and doctors, and tried to access their main source of information, if any, regarding HPV infection, transmission, and its prevention. Apparently, the title suggests that the authors tried to access the role of Internet-based awareness on HPV-related information and its impact on the six different cohorts through a cross-sectional study.

Specific comments:

  1. Title: The title of the manuscript is not focused on the actual study conducted by the authors, hence ambiguous and misleading. It suggests that the prime focus of this study is to identify the main source of HPV-related information, and Internet is the main game player, which is not completely true. Therefore, the title needs to be revised and simplified.
  2. What are the parameters fixed by the authors to judge the authenticity of the respondents' answers to the question “….. describe your level of knowledge on HPV infection”? Is there any separate questionnaire to determine the level of HPV-related knowledge among the cohorts?
  3. There are 7 options for answering the question of preventive measures for HPV infection. Few of the options are true to lower your chances of getting HPV. Is there any choice given to the respondents to select more than one option as their answer?
  4. There are multiple typological and/or grammatical errors identified throughout the text that need to be correctly proofread.

Author Response

Answer: We thank the reviewer for the comments.

  1. We ammended the title.
  2. The level of knowledge on HPV infection and vaccination was quantitatively assessed using a scale from 1 (very poor) to 5 (very good), from which the respondents could choose a single option. The following questions were answered by all respondents: “On a scale from 1 to 5, how do you characterize your level of knowledge on HPV infection?”, “On a scale of 1 to 5 how do you characterize your level of knowledge about HPV vaccination?”.
  3. The questions we used in our survey were open-ended, closed with ordered answers, closed with unordered answers, but also binary questions, either single or multiple choice. The question regarding which methods can be used to prevent HPV infection was a multiple choice question.
  4. We ammended the text.

Reviewer 2 Report

The work presents the extremely important problem of education related to HPV virus and the possibility of prophylaxis.

As the authors mention, the virus is one of the most common sexually transmitted infective agents and is associated with the risk of developing a number of cancers. It is worth adding other hpv-related cancers in the introduction, but at the same time note that not every infection leads to the development of a neoplasm. The virus can be eliminated by the host, the process of neoplastic transformation takes many years and cocarcinogens are also important.

A brief division of hpv into high- and low-risk types would be needed in the context of vaccines.

In the limitations section, mention should be made of the assessment for only one country and the subjectivity of the results. The questions did not verify the detailed knowledge about HPV, but asked for a subjectivw assessment.

In the discussion it is recommended to refer to data from other European countries, i.e. Poland (Parents' Knowledge and Attitude towards HPV and HPV Vaccination in Poland Katarzyna Smolarczyk et al. Vaccines (Basel). 2022. OR 

Assessment of the State of Knowledge about HPV Infection and HPV Vaccination among Polish Resident Doctors.

Smolarczyk K, et al. Int J Environ Res Public Health. 2021. PMID: 33440750)

Author Response

Answer: We thank the reviewer for the comments. We added supplementary information in the Introduction (Please see Introduction, Page 2, Lines 45-54, red colour) and the Discussion part (Please see Discussion, page 7, Lines 240-247, red colour; Limitations, Page 8, Lines 289-303, red colour). We introduced the suggested references in the article.

Reviewer 3 Report

The title of the manuscript looks too complex, and it impacts on its comprehension. If possible, it should be rephrased.  The abstract reflects the manuscript content very well.

The introduction part is too narrow and does not give a full understanding of the study rationale. Why do you focus only on cervical cancer? HPV causes a wide variety of conditions/cancers. And the vaccination prevents from all of them. Some data on the issues appeared with the national HPV vaccination program could be relevant. And what was the reason(s) of the HPV vaccination failure in Romania? That reason(s) could be linked to the subject of your current study.

The authors should also explain why they focus on the Internet as a source of information.  Any previous studies proved it to be the best? Suggested reference     https://doi.org/10.1080/21645515.2019.1581543.

Before stating the aim of the study in lines 69-71, please provide a data on how the knowledge on HPV, CC and vaccination could impact the situation. Suggested relevant references: doi: 10.1371/journal.pone.0261203; https:// doi.org/10.3390/vaccines10050824

In a nutshell, the existing text of the introduction should be restructured and expanded. The following structure for the introduction part is suggested:

1. HPV, HPV as a cause of various pathologic conditions.

2. Epidemiology of HPV in Romania

3. Epidemiology of CC and other HPV related conditions in Romania (if available)

4. Screenings and vaccination

4.1 Vaccination issues and its reasons

4.2 How knowledge on HPV and cancers affected the vaccination program

5. Sources for HPV knowledge

5.1 Internet as a source

6. Aims of the study.

The methods part should be restructured. Please mention the study design and then the study subjects. 

1. study design,

2. study subjects

3 study instruments

4 ethics

5. statistical analysis (more details required, which tests of stat analysis were applied). If the data analysis included descriptive statistics consisting, standard deviations, and frequencies? Independent variables? Chi-square test or Fisher’s exact test? Ordinal logistic regression? 

The results' section is supported by good tables, however a figure or two could improve the data comprehension.

In the discussion part, please report your study strengths and limitations.

And compare your study results with other countries with other developing countries with the same HPV vaccination experience (middle income). Suggested - 

Author Response

Answer: We thank the reviewer for the useful comments, which helped us make important modifications that improved our manuscript. However, we could not configure a comprehensive enough figure and remained with just the self-explanatory tables. We only focused on cervical cancer because of its high prevalence and mortality rates in Romania. We believe that only by raising awareness regarding the preventative role of HPV vaccine can we start making a difference. We ammended the title. We added supplementary data, as requested (Please see Introduction, Page 2, Lines 45-54, 70-74, Lines 78-114, red colour; Materials and Methods, Page 4, Lines 116, 120-122, 151-155, red colour; Limitations, Page 8, Lines 288-302, red colour).

Round 2

Reviewer 3 Report

Dear Authors, Thank you for the corrections made.